# Comparative Analysis of *SLA-1, SLA-2,* and *DQB1* Genetic Diversity in Locally-Adapted Kenyan Pigs and Their Wild Relatives, Warthogs

**DOI:** 10.3390/vetsci8090180

**Published:** 2021-09-02

**Authors:** Eunice Magoma Machuka, Anne W. Thairu Muigai, Joshua Oluoch Amimo, Jean-Baka Domelevo Entfellner, Isaac Lekolool, Edward Okoth Abworo, Roger Pelle

**Affiliations:** 1Biosciences Eastern and Central Africa, International Livestock Research Institute (BecA-ILRI) Hub, Nairobi P.O. Box 30709-00100, Kenya; J.DomelevoEntfellner@cgiar.org; 2Institute for Basic Sciences Technology and Innovation (PAUSTI), Pan African University, Nairobi P.O. Box 62000-00200, Kenya; 3Botany Department, Jomo Kenyatta University of Agriculture and Technology, Nairobi P.O. Box 62000-00200, Kenya; awmuigai@fsc.jkuat.ac.ke; 4Center for Food Animal Health, Department of Animal Sciences, 1680 Madison Avenue, The Ohio State University, Wooster, OH 44691, USA; amimo.3@osu.edu; 5Kenya Wildlife Services, Nairobi P.O. Box 40241-00100, Kenya; lekolool@kws.go.ke; 6Animal and Human Health Program, International Livestock Research Institute, Nairobi P.O. Box 30709-00100, Kenya; E.Okoth@cgiar.org

**Keywords:** SLA, diversity, domestic pigs, warthogs, MHC, locally-adapted pigs, DQB1, PBR

## Abstract

Swine leukocyte antigen (SLA) plays a central role in controlling the immune response by discriminating self and foreign antigens and initiating an immune response. Studies on SLA polymorphism have demonstrated associations between SLA allelic variants, immune response, and disease resistance. The SLA polymorphism is due to host-pathogen co-evolution resulting in improved adaptation to diverse environments making SLA a crucial genomic region for comparative diversity studies. Although locally-adapted African pigs have small body sizes, they possess increased resilience under harsh environmental conditions and robust immune systems with reported tolerance to some diseases, including African swine fever. However, data on the SLA diversity in these pigs are not available. We characterized the SLA of unrelated locally-adapted domestic pigs from Homa Bay, Kenya, alongside exotic pigs and warthogs. We undertook SLA comparative diversity of the functionally expressed SLA class I (*SLA-1*, *SLA-2*) and II (*DQB1*) repertoires in these three suids using the reverse transcription polymerase chain reaction (RT-PCR) sequence-based typing (SBT) method. Our data revealed higher genetic diversity in the locally-adapted pigs and warthogs compared to the exotic pigs. The nucleotide substitution rates were higher in the peptide-binding regions of the *SLA-1*, *SLA-2*, and *DQB1* loci, indicative of adaptive evolution. We obtained high allele frequencies in the three SLA loci, including some breed-specific private alleles, which could guide breeders to increase their frequency through selection if confirmed to be associated with enhanced resilience. Our study contributes to the growing body of knowledge on genetic diversity in free-ranging animal populations in their natural environment, availing the first DQB1 gene data from locally-adapted Kenyan pigs.

## 1. Introduction

The major histocompatibility complex (MHC) proteins play a vital role in binding and presenting endogenous and exogenous epitopes derived from the host or pathogen to the circulating T-cells, initiating a T-cell-mediated immune response [1,2,3,4,5]. MHC structure and functions have been widely studied in different species, including comparative MHC genetic diversity analyses in humans and the *Suidae* family that comprises *Sus scrofa* (wild boars), *Sus scrofa domesticus* (domestic pig), and warthogs (*Phacochoerus africanus*) [2,6,7,8]. However, no comparative MHC data are available from locally-adapted Kenyan pigs. 

In swine, MHC encodes the swine leukocyte antigen (SLA) mapped to the pig’s chromosome 7 [9,10]. SLA consists of three major gene families known as SLA class I, II, and III [9,11]. SLA class I genes comprise three classes: Classical (or class Ia) genes, non-classical (or class Ib), and MHC class I chain-related (MIC) genes (or class Ic). Classical MHC I molecules are highly polymorphic and present antigenic peptide ligands on infected cells to CD8^+^ T cells, whereas the non-classical MHC I molecules facilitate repressively or activate stimuli in natural killer cells [12]. Seven classical SLA class, I loci are known: *SLA-1*, *SLA-2*, *SLA-3*, *SLA-4*, *SLA-5*, *SLA-9*, and *SLA-11*, of which only *SLA-1*, *SLA-2*, and *SLA-3* are highly polymorphic and functional, with a classical MHC class I molecular structure consisting of a leader sequence, three exons encoding the extracellular α1, α2, and α3 domains, a transmembrane region, and three cytoplasmic exons [13,14]. The α1 and α2 domains form an 8- to 10-mer peptide-binding region (PBR). In pigs, a cluster of three non-classical class Ib genes consists of *SLA-6*, *SLA-7*, and *SLA-8* [9,13,14]. Conversely, SLA class II consists of *DMA*, *DMB*, *DOA*, *DOB1*, *DQA*, *DQB1*, *DRA*, *DRB1*, *DRB2*, *DRB3*, *DRB4*, and *DRB5* genes of which DRB1 and *DQB1* are highly polymorphic [2]. SLA class II is a heterodimer consisting of an alpha (α) and a beta (β) chain, both with an intracellular, a transmembrane, and two extracellular domains [13,15]. Pairing of the α1 and β1 domains from MHC class II form the PBR with open ends, allowing the apportionment for a larger peptide of 14-mer or more to extend out of both sides of the groove. The PBRs are highly polymorphic and diverse regions of the SLA, thereby ensuring greater diversity for binding and presenting peptides [13,16,17,18].

Genetic diversity in the host and pathogen populations is thought to be maintained by antagonistic co-evolution between interacting hosts and pathogens [19,20,21,22,23,24]. Comparative studies in the model and non-model organisms have shown that the high levels of genetic diversity observed at the MHC of vertebrate hosts are consistent with the hypothesis of pathogen-driven balancing selection through one of two mechanisms: Frequency-dependent selection and heterozygote advantage [18,25,26,27]. Consequently, high variations at the MHC loci reflect the level of fitness that can be exploited to genetically characterize different individuals in a population [18,28,29,30,31,32]. Thus, studying the variable sites of SLA molecules in as many suid breeds as possible is important to obtain crucial information on the SLA structure and the associations between SLA alleles and valuable traits. 

Pigs (*Sus scrofa domesticus*) are a source of high-quality protein, are easy to manage with minimal space and feed requirements [33,34,35], providing a livelihood source for smallholder farmers [36,37]. Pigs are also used as experimental models in biomedical research [38,39]. To this end, very few functional SLA class I and II diversity studies have been conducted on the domestic pigs and warthogs in East Africa [6,40]. The locally-adapted pig breeds are characterized by a rich genetic reservoir that has made them adaptive to harsh environmental conditions and disease challenges [41,42,43]. These pigs, found in diverse parts of Africa, show resilience under harsh climatic conditions [43] and tolerance to diseases [44,45,46] such as the lethal African swine fever (ASF) while remaining asymptomatic despite harboring the virus in their tissues [47,48,49,50]. The locally-adapted pigs are preferred by smallholder farmers in Africa who manage them in free-range systems (Figure 1B), exposing them to various pathogens and parasites. However, these locally-adapted pigs have a slower growth rate, resulting in smaller body sizes yielding lesser carcass weight than the exotic pigs such as Large white and Duroc breeds [51]. Despite this, these pigs are preferred by farmers who keep them for their resilience under harsh environments and disease challenges. We included exotic pigs (Landrace x Large white) that rapidly grow into big body sizes that yield higher carcass weight but are highly susceptible to many diseases, including ASF. Comparative genetic diversity was achieved by including the east African warthogs (*Phacochoerus africanus*) that live in the savannah areas at the livestock-wildlife interphase, where they play a vital role in the disease transmission cycle by acting as the reservoir for many pathogens, including ASF virus [52,53,54,55]. Warthogs grow at a slow rate and have small body sizes. The locally-adapted pigs, exotic pigs, and warthogs are closely related suid species but of different demographic histories, disease susceptibilities, and varying growth rates that make them a perfect combination to study the genetic diversity of SLA.

We analyzed the highly polymorphic and functionally expressed MHC class I (*SLA-1* and *SLA-2*) and class II (*DQB1*) loci of unrelated locally-adapted pigs from Homa Bay County in Kenya that have been shown to tolerate ASF for more extended periods against the natural ASF-resistant swine relatives, the warthogs [54,55,56], and a highly ASF-susceptible exotic domestic pig breed. This study supplements our understanding of the SLA diversity among the domestic pigs and their wild suid relatives, the warthogs. 

## 2. Materials and Methods

### 2.1. Sampling Sites 

The locally-adapted pigs (*Sus scrofa domesticus*) (Figure 1A) were randomly selected from smallholder farms in Homa Bay County, south-western Kenya. Lake Victoria borders Homa Bay County on the west, and the Ruma National Park is in the central part of this county (formerly Nyanza Province). The climatic conditions are warm and overcast most of the year, with temperatures ranging between 18–29 °C. The exotic pigs (*Sus scrofa domesticus*) (Figure 1B) were crosses of Large white and Landrace pigs randomly selected from a commercial pig farm called Farmer’s Choice Limited, located in Limuru Municipality, Kiambu County in central Kenya. The climatic conditions in Limuru are mostly cool and overcast most of the year, with temperatures ranging between 8–25 °C. The warthogs (*Phacochoerus africanus*) (Figure 1C) represent the wild suid population, collected from a national park in Garissa County, Kenya, at the far north-eastern side bordering the Tana River (Figure 1A). Garissa is an arid area characterized by mostly dry weather and temperatures ranging between 22–38 °C and rainfall in two seasons in February to April and October to December. The distance between Homa Bay and Limuru is about 273.35 km, Garissa to Limuru is 345.4 km, while Garissa and Homa Bay are 590.4 km apart (Figure 1D). Homa Bay County is an area endemic for ASF, while Limuru in Kiambu County experiences sporadic outbreaks of ASF.

### 2.2. Sample Collection 

Between 2014 and 2019, we collected blood from locally-adapted pigs, exotic pigs, and warthogs. Porcine (*Sus scrofa domestica*) blood was collected from the jugular vein into Vacutainers^®^ (Becton, Dickinson and Company, Franklin Lakes, NJ, USA) from 27 locally-adapted pigs (Figure 1A) from select farms in Homa Bay County, Kenya. Additionally sampled were 16 Landrace x Large White crossed exotic pigs, maintained by the Farmer’s Choice in Limuru, Kiambu County, Kenya. In addition, the study included 16 warthogs (*Phacochoerus africanus*) blood samples representing the wild suid population, collected from Garissa County, Kenya. Blood samples were collected from immobilized free-range warthogs. Blood samples were stored in RNA*Later*^®^ (Ambion, Austin, TX, United States) for RNA extraction.

### 2.3. RNA Extraction 

Total RNA was extracted either from 200 µL of whole blood using an in-house protocol that combined lysis and phase separation with 1 mL of TRIzol™ (Thermo Fisher Scientific, Waltham, MA, USA) and 200 µL Chloroform (Sigma-Aldrich, St. Louis, MO, USA) followed by on-column purification using the RNeasy^®^ Mini Kit (Qiagen, Hilden, Germany) following the manufacturer’s instructions. The RNA was eluted from the silica columns in 40 μL of RNase-free water. The RNA integrity was checked on a 1.2% (*w/v*) agarose gel electrophoresis at 4 V/cm^2^, and the RNA quantity was determined using an ssRNA assay kit on the Qubit^®^ 2.0 fluorometer (Thermo Fisher Scientific, USA) system. The RNA purity was determined on a Nanodrop™ spectrophotometer (Thermo Fisher Scientific, USA), considering the 260/280 ratio close to 2.0 as highly pure RNA. The extracted RNA was stored at −80 °C until required for downstream applications.

### 2.4. Reverse Transcription and PCR Amplification of SLA-1, SLA-2, and DQB1 RNA

The cDNA was prepared from 1 μg of the purified total RNA using the RevertAid First Strand cDNA Synthesis Kit (Catalog No. K1622, Thermo Fisher Scientific, USA), with random hexamers contained in the kit and following the manufacturer’s instructions. A GAPDH control was prepared alongside the cDNA reactions. Each cDNA sample was prepared in duplicate and used as a template for the SLA genotyping PCR. The full-length locus-specific primers for the *SLA-2* and *DQB1* loci were designed to include the untranslated regions to obtain the complete coding sequence [3]. For *SLA-1*, we designed primers to include only the alpha-1 region (exon 2) of the *SLA-1* gene. The SLA loci-specific PCR reactions were conducted under optimized conditions with 2U of Phusion™ Taq DNA polymerase (Thermo Fisher Scientific, USA) in 30 µL reaction volumes containing GC buffer, 2 mM of MgCl_2_, 0.2 mM of each dNTPs, 0.3 µM of each primer, and cDNA (prediluted 1:100 with nuclease-free water) as a template. The PCR thermal profile included an initial denaturation at 98 °C for 2 min, 35 cycles of 98 °C for 30 s, annealing at loci-specific temperatures (see Appendix A) for 30 s, and 72 °C for 30 s, with a final extension at 72 °C for 30 s. The amplicons were analyzed on a 1.5% (*w/v*) agarose gel pre-stained with 2.5 × GelRed^®^ (Biotium, Fremont, CA, USA) in 1 × TAE buffer and electrophoresed at 4 V/cm^2^ to confirm the amplicon size before purification with the QIAquick^®^ PCR Purification Kit (Qiagen, Germany). The cleaned SLA amplicons were sequenced in forward and reverse directions using the BigDye^®^ Terminator v3.1 cycle sequencing kit (Thermo Fisher Scientific, USA) on a 3730xl DNA Analyzer (Applied Biosystems, Waltham, MA, USA) at Macrogen Inc., Amsterdam, The Netherlands. The same SLA gene-specific primers for PCR amplification were used for sequencing, except *SLA-1* primers had universal T7 and SP6 tags for sequencing (Appendix A). We obtained clear amplicons of the expected sizes on agarose gels for the three loci studied. It was not possible to obtain a clear amplicon for sequencing in some loci, probably due to similarities in the primer-binding sites between the SLA class I genes in different pig breeds [40].

### 2.5. SLA Sanger Sequence Data Analysis and Construction of Phylogenetic Trees

The SLA class I and II gene-specific Sanger sequences were trimmed and assembled using the CLC Genomics Workbench v8.03 (Qiagen, Germany) with the default settings. Secondary peaks were detected to discover heterozygous mutations using the secondary peak calling plugin in CLC set at a 50% peak height threshold, a process augmented by visual inspection of the double peaks and manual editing. We disentangled the mixed signal represented by secondary peaks in Sanger chromatograms by reconstructing haplotype phases from the unphased sequence data containing ambiguous codes representing heterogeneous positions in PHASE v2.1.1 [57] at 100 burn-in steps followed by 100 actual iterations, each with a thinning interval of 1. The unfolded consensus sequences were used for sequence similarity search in BLASTn [58] in the NCBI database. The unfolded *SLA-1*, *SLA-2,* and *DQB1* nucleotide sequences were compared with the published SLA alleles in the IPD-MHC sequence database (http://www.ebi.ac.uk/ipd/mhc/sla/, accessed on 12 July 2021) to obtain the matching alleles [59]. Then, we constructed phylogenetic trees for two purposes: (i) To assign corresponding SLA alleles, and (ii) to infer orthologous relationships of the *SLA-1, SLA-2,* and *DQB1* genes within the MHC suids studied. Maximum likelihood (ML) phylogenetic trees were constructed in RAxML-NG v9.0 at https://github.com/amkozlov/raxml-ng (accessed on 12 July 2021) [60] with the GTR+G substitution model and edited in *iToL* [61]. The unfolded DNA sequences were further translated into amino acids using the *Translate* tool at Expasy Swiss Bioinformatics Resource Portal [62]. 

### 2.6. MHC-SLA Diversity Assessment

We used *hierfstat* package v0.5–7 under R v4.0.4 to estimate these genetic diversity parameters: Mean observed heterozygosity (Ho), mean gene diversity within the population (expected heterozygosity), total gene diversity (Ht), gene diversity among samples (Dst), heterozygosity corrected (Dstp), Fis = inbreeding coefficient per overall loci (Fis: 1-Ho / Hs); Dest = measure of population differentiation. Comparisons were made for each suid and all combined into one population for each of the three loci.

The multiple sequence alignments generated above were used to assess the variation between the unfolded SLA-specific sequences in DnaSP v6.0 [63]. The nucleotide diversity (π), haplotype diversity (Hd) Ewenns-Watterson neutrality test, and the total number of mutations per nucleotide site were also calculated for each individual in DnaSP v6.0 [63]. The PBR sequence diversity of each SLA loci-specific DNA sequence was analyzed in DnaSP v6.0 using default settings. 

### 2.7. Evaluating Signals of Selection Acting on the SLA-1, SLA-2, and DQB1 loci

To determine the diversity, conservation, and evolutionary forces acting within the selected MHC genes between domestic pigs and warthogs, we tested for selection in (i) the entire *SLA-1*, *SLA-2*, and *DQB1* codons and (ii) exons 2 to 3, the codons for the PBR in *SLA-2* and exon 2, the codons for the PBR in *DQB1.* Due to the high numbers of heterozygous positions within the loci studied, phased data were used for the selection analysis. The non-synonymous (dN) and synonymous (dS) substitutions rates were tested within each *SLA-1*, *SLA-2,* and *DQB1* loci’s coding regions using *CodeML* [64], a package within PAML v4.10.0, over the entire coding sequence and exclusively on the concatenated codons for the PBR, in the three suid populations (warthogs, local, and exotic pigs). We performed a Likelihood-Ratio Test (LRT) to compare three models: M0, M7 (beta), and M8 (beta and ω), a model including at least one category of sites under positive selection (ω = dN/dS > 1). The value obtained was used in the LRT at 2 degrees of freedom and allowed the rejection of the null model (*p* < 0.05). Amino acids detected to be under positive selection in M8 were identified using the Bayes Empirical Bayes approach (BEB), with posterior probabilities of 95% and 99% [64]. The amino acid positions were numbered according to the published sequences in GenBank. For *SLA-1*, we assigned amino acid positions by comparing the sequences against CAB63936.1, the genomic DNA translation of exon 2 of AJ251829 (*SLA-1*01:01*). For *SLA-2*, we used AJ251829 that corresponds to the *SLA-2*03:01* allele, and for *DQB1*, we compared against NM_001113694.1, the mRNA sequence encoding *SLA-DQB1*03:01* allele.

### 2.8. SLA Functional Cluster Analysis

Identifying the peptides that bind to the MHC is critical to understanding the T-cell immune responses. We performed the MHC cluster analysis to assess the clustering of the functional SLA peptide-binding motifs against known peptides. The phased *SLA-1, SLA-2,* and *DQB1* nucleotide sequences were first translated to proteins. The longest open reading frames (ORFs) alongside protein sequences from the IPD-MHC database were selected and submitted for the MHC cluster analysis on *MHCcluster* 2.0 (http://www.cbs.dtu.dk/services/MHCcluster/; accessed on 20 September 2020) [65] by comparing the peptide-binding function of 50,000 random natural 10 amino acid long peptides of known SLA alleles. Default parameters were applied, and each SLA locus was analyzed separately. 

### 2.9. Statistical Analyses

The allele frequencies were determined by direct counting, along with their respective standard errors. We applied the likelihood-ratio test (LRT) to compare three substitution rate models: M0, M7 (beta), and M8. The value obtained was used in the LRT at 2 degrees of freedom and rejected the null model (*p* < 0.05). Amino acids under positive selection in M8 were identified using the BEB approach, with posterior probabilities of 95% and 99% [66]. For the MHC cluster analysis, we used 100 bootstrap samples, and 50,000 peptides in the functional correlation analysis, at a threshold of 10 peptides. The estimated accuracy of the predicted sequence motifs was estimated from the distance to the nearest MHC molecules included in the training of the peptide binding prediction method. Correlations > 0.7 were considered accurate.

## 3. Results

### 3.1. SLA Sequence Analysis of Domestic Pigs and Warthogs 

RT-PCR amplification was performed on blood samples from 59 animals (43 domestic pigs and 16 warthogs) using the locus-specific MHC SLA primers (Appendix A). The average sequence sizes for *SLA-1*, *SLA-2,* and *DQB1* loci were 240, 1056, and 917 bp, respectively, after assembly of the forward and the reverse sequences and trimming off the T7 and SP6 tags on *SLA-1* primers (Appendix A). The secondary signals in the Sanger sequences with assigned IUPAC ambiguous codes were resolved by reconstructing haplotype phases in PHASE v2.1.1 yielding at least two haplotypes per heterozygous sample. A BLAST search showed similarity to known *Sus scrofa* MHC class I antigens (*SLA-1* and *SLA-2*) and MHC class II antigens (*DQB1*) mRNA (Appendix A). Using the standard genetic code, the longest ORF for all the *SLA-1*, *SLA-2*, and *DQB1* nucleotide sequences yielded, on average, 80, 365, and 271 (amino acids) aa, respectively (Appendix A). 

### 3.2. Phylogenetic Analysis

Phylogenetic relationships were determined using ML trees from *SLA-1*, *SLA-2,* and *DQB1* phased nucleotide sequences (Figure 2, Figure 3 and Figure 4). The ML trees showed clustering based on allele nomenclature, verifying further that correct allele names were assigned. Allele sharing among suid breeds was detected, with some alleles clustering separate from the others due to differences in the length of the nucleotide sequences used to generate the phylogenetic trees. 

### 3.3. SLA-1, SLA-2, and DQB1 Allele Assignment 

We analyzed the *SLA-1*, *SLA-2*, and *DQB1* phased nucleotide sequences from 60 individuals (locally-adapted pigs n = 27, exotic pigs n = 16, and warthogs n = 16) on SLA-IPD BLAST and selected allele names with >95% match (Appendix A). We obtained 28 unique *SLA-1* alleles in this study (Figure 5; Appendix A). The most frequently detected *SLA-1* allele was *SLA1*07:07* (20%) in all three suids studied. The second most frequent *SLA-1* allele was *SLA1*14:04* (15.79%) and the third being *SLA1*14:05* (8.42%) detected in all three suids. The rest of the *SLA-1* alleles occurred at frequencies of less than 5.6%. Minor alleles were detected at frequencies less than 1% (Figure 5; Appendix A). A total of 17 private *SLA-1* alleles were detected among the three suids. Of these private alleles, 11, 3, and 3 were for locally-adapted pigs, exotic pigs, and warthogs, respectively. The high frequency (>5%) alleles were shared by all three suids detected namely *SLA-1*07:07, SLA-1*08:18, SLA-1*14:04, SLA-1*14:05,* and *SLA-1*18:01.* The local and exotic pigs shared alleles *SLA-1*09:01, SLA-1*11:06,* and *SLA-1*16:03.* The locally-adapted pigs and warthogs shared alleles *SLA-1*08:16* and *SLA-1*14:02.* The exotic pigs did not share any *SLA*-1 alleles with the warthogs (Appendix A).

A total of 21 unique *SLA-2* alleles were obtained in this study (Figure 5; Appendix A). The most frequently detected *SLA-2* allele was *SLA-2*06:07* (19.64%) in all three suids studied. Other high-frequency *SLA-2* alleles were *SLA-2*05:04* and *SLA-2*04:02:02,* occurring at a frequency of 10.71%. The rest of the *SLA-2* alleles occurred at frequencies of less than 5%. Minor alleles were detected with frequencies of 1.79% (Figure 5; Appendix A). A total of 15 private *SLA-2* alleles were detected among the domestic pigs only. Of these private alleles, seven belonged to locally-adapted pigs and eight were from exotic pigs. The local and exotic pigs shared five alleles namely *SLA-2*02:02, SLA-2*04:02:02, SLA-2*12:01, SLA-2*02:01,* and *SLA-2*10:09.* The locally-adapted pigs and warthogs shared only one allele *SLA-2*06:07.* The exotic pigs did not share any *SLA-2* alleles with the warthogs (Appendix A).

There were 11 unique SLA-*DQB1* alleles obtained in this study (Figure 4; Appendix A). The high-frequency *DQB1* allele was *SLA-DQB1*07:01:02* (52.38%) detected in all three suids, followed by *SLA-DQB1*07:01:01* (21.43%). While *SLA-DQB1*02:02* and *SLA-DQB1*08:04* were detected at a frequency of 4.76%. Minor alleles unique to each suid breed were detected with frequencies of 2.38% each, comprising the nine private *DQB1* alleles detected in the domestic pigs (Figure 4; Appendix A). Of these *DQB1* private alleles, three belonged to locally-adapted pigs and six were from exotic pigs. The local and exotic pigs shared one allele with the warthogs, the *SLA-DQB1*07:01:02* (Appendix A).

### 3.4. Selection Signals Acting on SLA-1, SLA-2, and DQB1 Loci

Following the selection analyses, the estimated ω value under the M8 model for only the sites under positive selection in the locally-adapted pigs and warthogs was higher (ω > 1; *p* < 0.05) in the loci studied in the three suids. The ω = dN/dS value under the M8 model was significantly higher (ω > 1; *p* < 0.05) in all the PBR regions of *SLA-1, SLA-2,* and *DQB1* in all the three suids studied (Table 1, Table 2 and Table 3). When the (overall) global M0 model was considered, our results showed that the ω value was significantly less than 1 (ω < 1; *p* < 0.05) when all coding sequences and individual coding regions (exons) for *SLA-1*, *SLA-2,* and *DQB1* were considered. Moreover, the *DQB1* loci ω value was significantly higher (ω > 1; *p* < 0.05) in the locally-adapted pigs when whole sequences of *DQB1* were considered (Table 3). However, other non-PBR exons also had higher ω values, but were not significant, namely: *DQB1* exon 4, *DQB1* exon 5 (in warthogs and exotic pigs only), *SLA-2* exon 6 in all sequences and exotic pigs, and lastly, *SLA-2* exon 7 among locally-adapted pigs had ω = 2 though not significant (Table 2). The warthog sequences available for the *SLA-2* and *DQB1* analysis were all identical.

We detected 10 positively selected codons in *SLA-1* loci when all suid breeds were considered, with eight being significantly positively selected sites at 0.95 and 0.99 probabilities. The exotic pigs alone had 22 sites, and the locally-adapted pigs had 19 sites (Appendix A). 

In all *SLA-2* codons, the total number of significantly positively selected sites at 0.95 and 0.99 posterior probabilities were 18, 30, and 31 in all suids, exotic pigs, and locally-adapted pigs, respectively (Appendix A). The available warthog sequences were strictly identical, so we did not perform within-group analyses for warthogs.

In *DQB1* loci, all codons combined, we detected a total of 27 positively selected sites when all suid breeds were considered. The exotic pigs alone had 27 sites, and the locally-adapted pigs had 34 sites, while the warthog sequences were not considered since the comparison between M7-M8 was not significant because they were strictly identical. In the *DQB1* loci, the number of significantly positively selected sites at 0.95 and 0.99 probabilities were 18, 18, and 11 in all suids, exotic pigs, and locally-adapted pigs, respectively (Table 4). Most of the positively selected codon sites within DQB1 were common across the three suid breeds studied, with a high proportion of positive selection occurring in the PBR-coding regions (Table 4; Appendix A). 

### 3.5. SLA-1, SLA-2, and DQB1 Genetic Polymorphism and Differentiation

For the *SLA-1* loci, a total of 60 individuals (16 exotic pigs, 27 local pigs, and 16 warthogs), each with 240 nucleotide sites were used to analyze the genetic diversity. Genetic diversity parameters and the results of neutrality tests are shown in Table 5. The observed heterozygosity (Ho) ranged between 0.0341 to 0.0778, while the expected heterozygosity was between 0.0887 to 0.1637, whereby the locally-adapted pigs and warthogs had higher Ho, Hs, and Ht values compared to the exotic pigs. The Fst values between populations ranged from 0 to 0.0485, and the Dst value was 0.0058 when all the individuals were combined into one. The inbreeding coefficient was highest among the exotic breeds (Fis = 0.6154). The measure of population differentiation (Dest) was 0.0087 when all the individuals were combined.

In *SLA-2* loci, a total of 43 individuals (16 exotic pigs, 25 local pigs, and two warthogs), each with 1056 nucleotide sites were used to analyze the genetic diversity (Table 5). The observed heterozygosity (Ho) ranged between 0.0038 to 0.0167, while the expected heterozygosity was between 0.0019 to 0.0.0633, whereby the locally-adapted pigs and exotic pigs had higher Ho, Hs, and Ht values compared to the warthogs in which the power of numbers were limiting. The Fst values between populations ranged from 0 to 0.173, and the Dst value was 0.0096 when all the individuals were combined into one. The inbreeding coefficient was higher among the locally-adapted pigs (Fis = 0.7367) than the exotic pig breeds, which had an inbreeding value of 0.653. The measure of population differentiation (Dest) was 0.0135 when all the individuals were combined.

In the *DQB-1* loci, a total of 39 individuals (16 exotic pigs, 21 local pigs, and two warthogs), each with 759 nucleotide sites were used to analyze the genetic diversity (Table 5). The observed heterozygosity (Ho) ranged between 0.0006 to 0.002, while the expected heterozygosity was between 0.0099 to 0.0197, whereby the exotic pigs and warthogs had higher Ho, Hs, and Ht values compared to the locally-adapted pigs in which the power of numbers were limiting. The Fst values between populations ranged from 0 to 0.287, and the Dst value was 0.0059 when all the individuals were combined into one. The inbreeding coefficient was higher among the locally-adapted pigs (Fis = 0.9428) than the exotic pig breeds, which had an inbreeding value of 0.8911 and warthogs at 0.8696. The measure of population differentiation (Dest) was 0.008 when all the individuals were combined.

The *SLA-1* sequences from the locally-adapted pigs and warthogs showed greater nucleotide diversity (π = 0.10083 ± 0.0053 and 0.11654 ± 0.00952) compared to those from the exotic pigs (0.085 ± 0.00796). The locally-adapted pigs had the lowest haplotype diversity at the PBR of *DQB1* loci (Table 6). The warthogs had a larger θW at exon 2 of the *SLA-1* loci among the three study groups. Theta per site from Eta (θ_Eta_) was considerably higher in warthog samples (θ_Eta_ = 0.166), followed closely by the locally-adapted pigs (θ_Eta_ = 0.13146) at the *SLA-1* loci. The exon 3 of the *SLA-2* loci nucleotide sequences from the exotic pigs showed greater nucleotide diversity (π = 0.06599 ± 0.00461) compared to those from the locally-adapted pigs (π = 0.05030 ± 0.00434). The warthogs had the lowest nucleotide diversity at the *SLA-2* exon 2 (Table 6). The locally-adapted pigs had higher θW values at the exon 2 of *SLA-2* loci while theta (θ_Eta_) was relatively higher in exotic pigs, followed closely by the locally-adapted pigs’ exon 3 of *SLA-2* loci. At exon 2 of the *DQB1* loci, the exotic pigs showed greater nucleotide diversity while Theta (θ_Eta_) was considerably higher in exon 2 of the *DQB1* loci exotic pigs than in the locally-adapted pigs (Table 5).

### 3.6. MHC Functional Cluster Analysis 

To determine MHC peptides function-based clustering, we selected the representatives of the *SLA-1*, *SLA-2*, and *DQB1* alleles from among the domestic pigs and warthogs for analysis using *MHCcluster* 2.0. The longest ORF for each SLA gene-specific loci was submitted for analysis on *MHCcluster* 2.0 [65] to allow for the comparison of the peptide-binding function of 50,000 random natural 10 amino acid long peptides of known SLA alleles and predict the binding motifs for the SLA Class I and II proteins [65]. We selected 29, 10, and 10 of the *SLA-1*, *SLA-2*, and *DQB1* as sequence representatives and added the corresponding MHC alleles in the *MHCcluster* 2.0 server. The peptide-binding motifs were visualized in the form of an unrooted tree and a heatmap for similarities of predicted binding to a set of predefined natural peptides. The analysis of *SLA-2* peptides generated 12 significant clusters (Figure 6A,B). The intensity of the color on the heatmaps correlates with the behavior of the epitopes’ predictive ability between different MHCs (Figure 6A,C). The red sections represent the regions where the alleles of MHC with a similar high affinity anchor to the epitopes, the yellow sections represent the regions where the similarity of affinity with epitopes is distant. The warthog polypeptide sequences included showed functional similarity to those of the locally-adapted pigs. The *SLA-2**05:01, *SLA-2**05:02, and *SLA-2**12:01 alleles clustered together on the heatmap and tree generated, while *SLA-2**03:02 and *SLA-2**10:02 also clustered together, indicating functional homologies, with accuracy levels of 1.00 (100% match), and that the molecules were characterized with high precision by the peptide-binding data.

The analysis of *DQB1* peptides generated seven major clusters (Figure 6C,D). The *DQB1* warthog polypeptide sequences included in *MHCcluster* analysis showed functional similarity to those of the locally-adapted pigs and exotic pigs. The SLA-*DQB1**07:01 and SLA-*DQB1**05:01 alleles clustered together, and SLA-*DQB1* *09:01 and SLA-*DQB1**08:01 alleles also clustered together, indicating functional homologies between alleles. The clustering was supported by accuracy levels of 1.00 (100%), indicating that the molecules in each cluster obtained were highly similar to the known peptide-binding motifs. The DRB1 sequences included in the *DQB1* analysis formed a distinct cluster (Figure 6C,D; dotted black lines) and were thus functionally unique. The *SLA-1* polypeptides generated in this study did not show similar or strong peptide-binding motifs (accuracy level < 0.5) with known *SLA-1* obtained from IPD and the *MHCcluster* server (Appendix A).

## 4. Discussion

Analyzing genetic diversity within MHC genes is useful in understanding their adaptive significance in host immune function towards improving pig health and guiding pig breeding strategies. This study aimed at comparing SLA class I (*SLA-1* and *SLA-2*) and class II (*DQB1)* genetic diversity for the first time in locally-adapted pigs and their wild relatives, warthogs from Kenya. Our results show greater MHC diversity in the locally-adapted pigs and warthogs previously reported to be more resilient under harsh climatic conditions with some levels of tolerance or resistance to disease [43,48] than the exotic pigs. The factors driving this MHC diversity are likely to be interactions between differential fitness and pathogenic pressures. 

Our study identified a high number of SLA class I (*SLA-1, SLA-2*) and class II (*DQB1*) sequence variants in the three suid populations. However, we refrain from categorizing these sequence variants as novel alleles since these MHC sequence variants were inferred using phased sequence data (see the Methods Section). Ultimately, the validation of alleles by resequencing of clones or re-genotyping of individuals with rare haplotypes is imperative.

### 4.1. Comparative Analysis of SLA Alleles and Sequence Diversity 

Genetic diversity is fundamental in biodiversity and adaptation of species and a high genetic diversity is an indicator of stronger viability and adaptability of populations to a given environment [67]. Our study found low Fst values in all three loci, indicating low genetic differentiation between populations, confirming frequent gene flow between populations, which increases the species’ adaptability to changes in the environment. We found higher genetic diversity amongst the locally-adapted pigs and warthogs at the *SLA-1* loci than the exotic pigs that undergo intensive inbreeding to maintain important reproductive traits for commercial purposes (Table 5). The higher diversity may enhance the tolerance of the locally-adapted pigs and warthogs populations to changes in the environment resulting in higher adaptability to novel environmental pressures [8,67].

In this study, we observed differences in MHC allele distribution between the exotic, locally-adapted pigs, and warthogs which could be attributed to their fitness level resulting from a natural or artificial selection over time. Our study identified 49 class I (*SLA-1*, *SLA-2*) and 11 class II (*DQB1)* alleles in the samples of the three suid populations studied. The locally-adapted pigs had a high number of alleles, although the allele distributions in the three loci were highly skewed, whereby at least one allele occurred at a higher proportion than others, consistent with previous MHC studies [16]. The skewed occurrence of alleles could be due to the adaptive evolution driven by natural selection to increase the frequencies of beneficial alleles and decrease frequencies of deleterious alleles [18,68].

Similar to previous studies, we reported a high proportion of *SLA-1*, *SLA-2*, and *DQB1* breed-specific alleles among the locally-adapted pigs and warthogs compared to the exotic pigs [6,8]. The alleles occurring at high frequency were detected amongst the locally-adapted pigs and warthogs. We found several *SLA-1* and *DQB1* alleles shared by the domestic pigs and warthogs, which occurred at high frequencies (Appendix A). The alleles shared by all three suids indicate the early origin of these alleles in suids and their possible importance in antigen recognition [8] as it is generally recognized that a high diversity of MHC alleles confers protection to a range of pathogens [6,22]. A large number of breed-specific private alleles (26, 29, and 11 for *SLA-1*, *SLA-2*, and *DQB1*, respectively) observed in this study indicate the impact of recent selective breeding or possible barriers to gene transfer among the sample of suids populations studied [3,8].

High allele sharing was observed at the *SLA-1* locus among the suid breeds, indicating convergence evolution at this locus that likely confers a fitness advantage to these suid breeds. *SLA-2* had low allele sharing among the three suid breeds studied, with a large number being shared between the domestic pigs only. At the *SLA-2* locus, only one allele, *SLA-2*06:07*, was shared between local pigs and the warthogs, occurring at a high frequency (19.64%) indicative of convergence evolution and maintenance of this allele during breed diversification [8]. We observed a different pattern in SLA class II (*DQB1*) genes, whereby a lower rate of breed-specific alleles was identified than the SLA class I genes, probably due to the fact that the SLA class II genes are less polymorphic than the SLA class I genes [8]. Thus, the SLA class I genes are better suited markers to study the history of the suid species than the SLA class II genes.

Our study reported several private SLA alleles in all three suid breeds studied (Appendix A). Private alleles are indicators of gene flow, where they can be correlated with the mean number of migrants exchanged per generation between populations. Private alleles unique to each pig breed are essential in screening potential T-cell epitopes for diagnostics and vaccine development [69,70,71]. In addition, the unique or private breed-specific alleles may guide pig breeders to selectively increase the frequency of the specific SLA alleles, primarily if shown to be associated with advantageous agronomic traits and disease tolerance or resistance. 

PBRs directly influence the peptides that the SLA allele can bind, and this information is essential in peptide-histocompatibility for peptide-based vaccines [69,70,71]. We compared the PBR-encoding sequences across the three suid breeds for each locus studied to infer orthologous relationships despite inherent phenotypic differences. We observed high SLA sequence polymorphisms within the PBR, as reported in previous studies [8,18,72,73]. PBR-encoding sequences of the east African warthogs grouped with locally-adapted pigs indicated high similarity between the SLA motifs. This could imply that the local pigs and warthogs have undergone similar host-pathogen co-evolutionary pressures [6,74,75]. The observed homology between the locally-adapted pigs and warthogs contradicts a previous study that showed that the locally-adapted Homa Bay pig populations did not share genetic similarities with wild pigs using the porcine SNP60/80 chip to genotype the pigs [43]. Our findings affirm SLA as a superior adaptive marker, capable of deciphering genetic diversity relevant to conservation compared to commonly used markers outside the MHC regions [18,76]. The SLA class II *DQB1* exon 3 PBR-encoding sequences did not yield any significant clustering pattern since the *DQB1* genes are more conserved than *SLA-1* and *SLA-2*, and therefore, intragenic variations would be difficult to decipher [77].

### 4.2. MHC Functional Cluster Analysis

The MHC peptide-binding groove contains polymorphic residues responsible for the different peptide specificities of the different alleles, ensuring that some individuals within a population can recognize protein antigens produced by virtually any microbe, reducing the likelihood that a single pathogen can evade host defenses in all individuals in a given species. When we compared the SLA sequences of alleles of the polymorphic class I and class II loci, we found that the nucleotide substitutions were concentrated in the exons that encode the peptide-binding groove, the site of interaction with the T-cell receptor [15,78]. It is crucial to characterize the peptide-binding motifs to identify peptides that respond directly to viral antigens to establish their adaptive immunity roles. The clustering of *SLA-2* alleles confirmed the presence of 12 specificity supertypes or specificity groups (Figure 6A,B) in our study cohort. The warthog *SLA-2* polypeptide sequences included in this cluster analysis showed functional similarity to those of the locally-adapted pigs (Figure 6A,B). We observed shared functional homology between the different *SLA-2* alleles illustrating the immunological phenotypic similarities between pig breeds, particularly those heterozygous at SLA class I loci [79]. Functional peptide clustering of the most frequent *DQB1* alleles in the study cohort yielded seven specificity supertypes or specificity groups (Figure 6C,D). The SLA-*DQB1* *07:01 and SLA-*DQB1* *05:01 alleles clustered together, and SLA-*DQB1* *09:01 and SLA-*DQB1* *08:01 alleles also clustered together, indicating functional homologies between these alleles. In this same cluster, the warthog *DQB1* peptide sequences (LR882540) showed functional similarity to peptide sequences from both the exotic and locally-adapted pigs (Figure 6A,B). Four unique clusters that contained peptide sequences from exotic pigs (LR882519 and LR882521) and two from locally-adapted pigs (LR882505 and LR882508) were obtained, demonstrating that these four pigs contained peptides unique at the *DQB1* loci.

Due to the smaller size of the predicted *SLA-1* protein in this study, we were not able to perform a cluster similarity analysis on the *MHCcluster* method, which has been shown to perform functional similarities at best between large sets of MHC molecules [65,79]. These peptides can be synthesized to induce and screen for the T-cell response, thus broadening the SLA repertoires and consequently attaining increased chances of matching animals in terms of SLA expression in future peptide–epitope studies [2,17,79].

### 4.3. Differences in SLA Genetic Diversity and Selection Analysis

MHC diversity is beneficial to the host when faced with emerging diseases due to its role in recognizing new pathogens and survival during infection. We observed differences in MHC diversity between the exotic pigs, locally-adapted pigs, and warthogs, which could be attributed to differences in exposure to pathogen communities. Pathogens have evolved to escape recognition by the most frequent SLA molecule, resulting in several MHC variants expressed by the host so that the pathogens do not totally evade detection [73]. These high levels of polymorphism of the MHC genes are maintained by balancing selection to increase the repertoire of antigen-presenting molecules on the cell surface [23]. The genetic signature of this positive selection results in an increased ratio between non-synonymous and synonymous substitution, particularly within the PBR-encoding regions but not outside the PBR region. An invaluable statistic to measure the strength and mode of natural selection acting on the protein-coding genes is (omega) ω = dN/ dS [80]. The ω value summarizes the evolutionary rates of genes and identifies conserved genes and those that have undergone periods of adaptive evolution. Our study obtained higher ω values (ω > 1) in all the loci studied and in the PBR-encoding regions, where the ω value was statistically significant (ω > 1; *p* < 0.05). The locally-adapted pigs and warthogs consistently had higher ω values (ω > 1) except in *DQB1,* where the exotic pigs had higher values than the locally-adapted pigs and warthogs, indicating that *DQB1* had undergone adaptive evolution among the exotic pigs studied. Thus, our analysis confirms a strong positive selection within the PBR-encoding regions of *SLA-1*, *SLA-2*, and *DQB1* involved in binding pathogen-derived epitopes consistent with previous MHC diversity studies [18,29,72,73,81]. However, the dN/dS ratio analyses are limited in explaining events that occur at shorter time scales, such as during a high pathogen load [18].

There were more site-specific codons being under positive selection in PBR regions or close to PBR codons, consistent with previous MHC studies [72,73], confirming that these residues have a critical role maintained by balancing selection in the suid immune function. We found that MHC class I, specifically *SLA-2* PBR-encoding regions, had a higher proportion of positively selected codons than *DQB1* (Table 4), indicative of exposure to epitopes of exogenic origin. Since the analysis of site-specific selection in warthogs was constrained by the small number of *SLA-2* and *DQB1* sequences obtained in our study, only *SLA-1* (exon 2; PBR) yielded site-specific selection data from warthogs in which five out of the 12 sites identified were significantly positively selected at posterior probabilities of 0.95 and 0.99 (Table 6). We found several similar positively selected codons among the three suid species studied (Table 5, Appendix A and Appendix A), indicating similar selective pressures acting on these codons. Likewise, the presence of several independent positively selected codons (Table 5) in these different suid species indicates that they may have functional implications on peptide binding and further contribute to the observed polymorphism [72,73].

Overall, our analysis showed positive balancing selection for the SLA regions studied, a distinctive feature of MHC genes that elevates the repertoire of antigen-presenting molecules on the cell surfaces, increasing the diversity of the antigen-presenting peptides, thus providing an advantage to the host in combating different infectious diseases [6,18,72]. Our findings support the hypothesis that pathogen-driven selection is the primary driver of MHC diversity [16]. The differences observed in diversity between the locally-adapted and exotic pigs can be attributed to the differential fitness of the swine hosts and differences in exposure to pathogens and parasites. Thus, the lower SLA diversity observed in the exotic pigs could be due to high selection and resultant inbreeding to maintain the desirable production traits in modern pig production [18]. However, the extent of SLA diversity reported in this study may not fully reveal that of the domestic Kenyan pigs and warthog populations due to the relatively small sample size, the possibility of founder effects in this population, the breeding schemes, and management systems that were not considered.

## 5. Conclusions

We undertook a comparative SLA genetic diversity analysis of three closely related suids: Exotic and locally-adapted Kenyan pigs and warthogs, reporting differences in the distribution of alleles among the suid breeds as well as breed-specific private alleles. Our study provides substantial evidence of positive selection acting on the PBR-coding region of the *SLA-1*, *SLA-2,* and *DQB1* genes in domestic pigs and warthogs. In addition, we identified several positively selected codons in the domestic pigs and warthogs that can be targeted for association studies with disease resistance or susceptibility traits. The findings from our study contribute to the body of knowledge on genetic diversity in free-ranging animal populations in their natural environment, availing the first *DQB1* gene data from locally-adapted Kenyan pigs. Therefore, we recommend extensively and carefully designed population-level and experimental challenge studies to determine the association between SLA diversity and valuable agronomic traits among the locally-adapted pigs and warthogs, which are crucial in pig genetic improvement programs.

## Figures and Tables

**Figure 1 vetsci-08-00180-f001:**
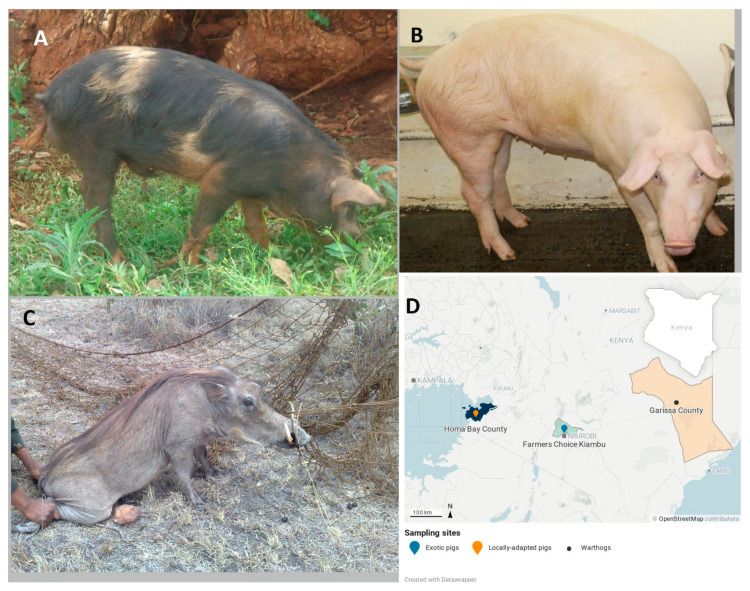
(**A**) A locally-adapted pig feeding at a homestead in Homa Bay County. Photograph: Edward Okoth/2016. (**B**) The exotic pig in a pen. *Photograph: Eunice Machuka/2019*. (**C**) An adult warthog being restrained in a net by a Kenya Wildlife Service Officer. *Photograph: Edward Okoth/2018*. (**D**) Map of Kenya showing the sampling areas in Homa Bay, Farmers Choice in Kiambu County, and Garissa county (Map was created using *Datawrapper*. https://www.datawrapper.de; accessed on 1 May 2021).

**Figure 2 vetsci-08-00180-f002:**
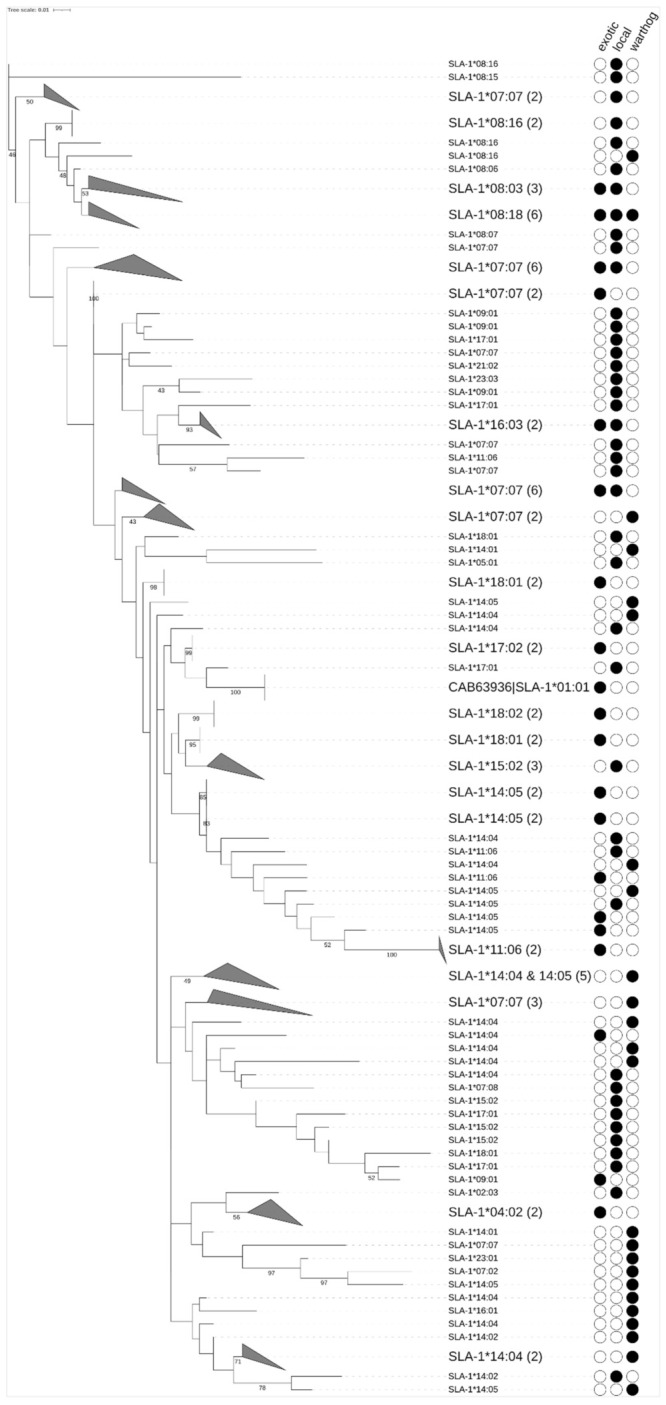
Phylogenetic tree of *SLA-1* alleles with leaf names, the closest match in IPD from a Blast search. Clades are collapsed (gray triangle) where the assignment to IPD alleles is the same (or very similar). Filled circles indicate the allele detected in the suid breeds studied. In parentheses are the number of phased haplotypes included in a collapsed node. Bootstrap support values were calculated on 100 bootstrap trees and values lower than 40 are not shown. CAB63936 is an *SLA-1* allele reference sequence obtained from GenBank.

**Figure 3 vetsci-08-00180-f003:**
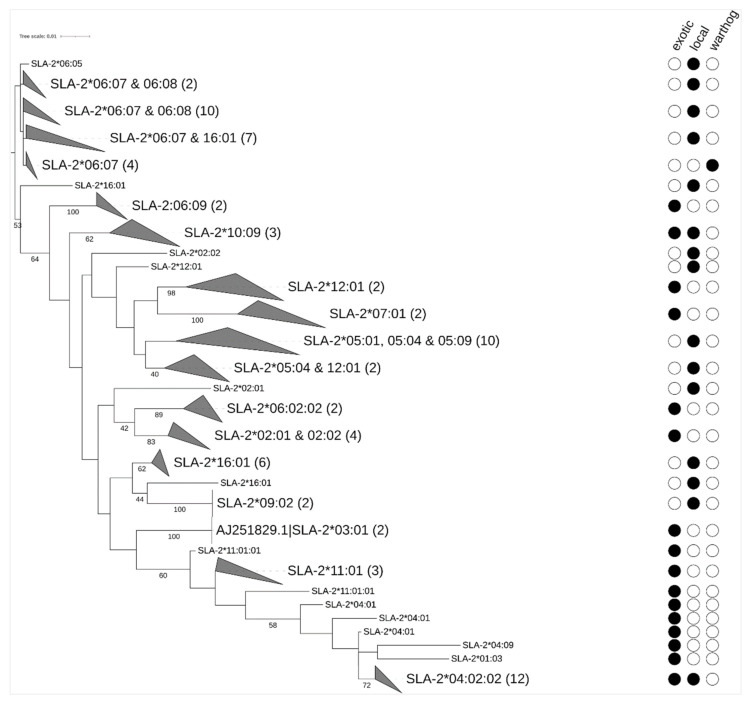
Phylogenetic tree of *SLA-2* alleles with leaf names, the closest match in IPD from a Blast search. Clades are collapsed (gray triangle) where the assignment to IPD alleles is the same (or very similar). Filled circles indicate the allele detected in the suid breeds studied. In parentheses are the number of phased haplotypes included in a collapsed node. Bootstrap support values were calculated on 100 bootstrap trees and values lower than 40 are not shown. AJ251829 is an *SLA-2* allele reference sequence obtained from GenBank.

**Figure 4 vetsci-08-00180-f004:**
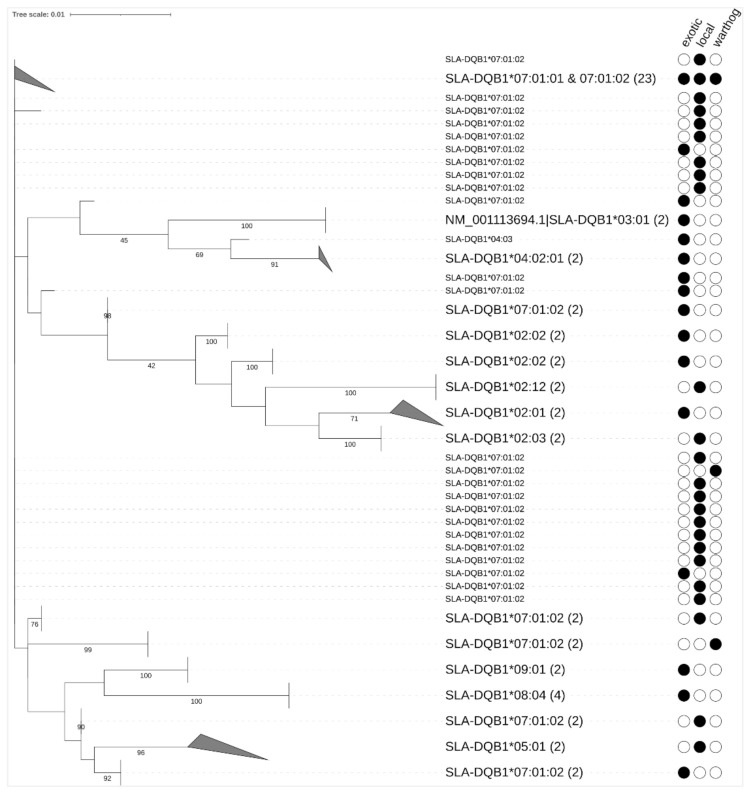
Phylogenetic tree of *DQB1* alleles with leaf names, the closest match in IPD from a Blast search. Clades are collapsed (gray triangle) where the assignment to IPD alleles is the same (or very similar). Filled circles indicate the allele detected in the suid breeds studied. In parentheses are the number of phased haplotypes included in a collapsed node. Bootstrap support values were calculated on 100 bootstrap trees and values lower than 40 are not shown. NM_001113694.1 is a *DQB1* allele reference sequence obtained from GenBank.

**Figure 5 vetsci-08-00180-f005:**
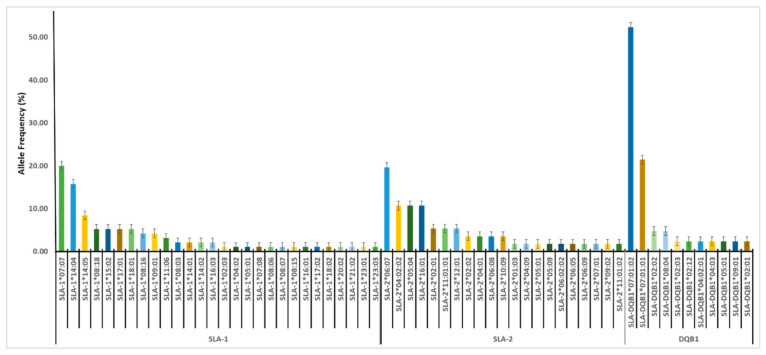
SLA allele frequencies grouped by SLA loci studied. Error bars indicate the standard error.

**Figure 6 vetsci-08-00180-f006:**
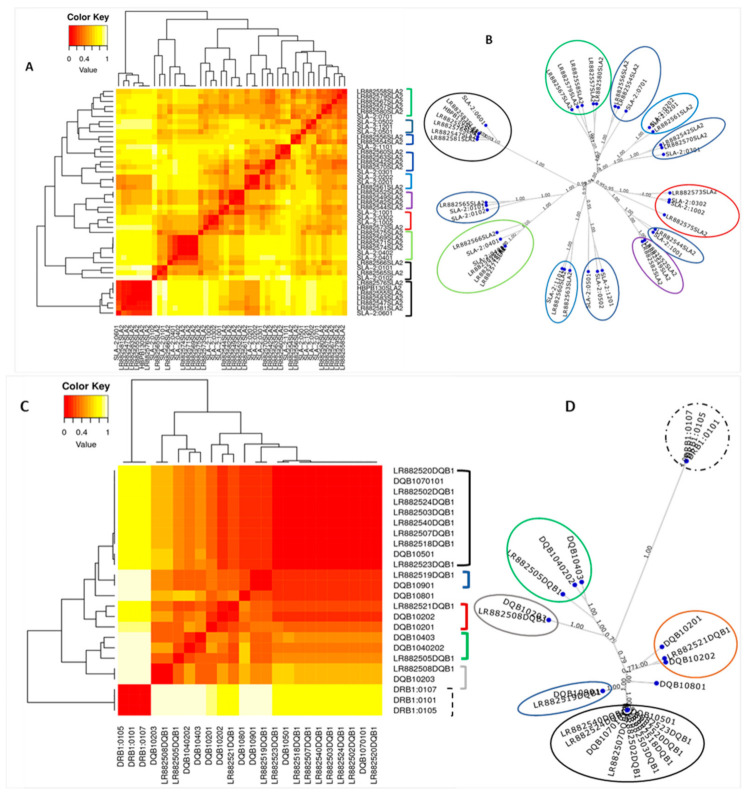
MHC functional cluster analysis of the peptide sequences of entire amino acid sequences of the SLA-2 and DQB1 representative alleles identified in this study from locally-adapted pigs, exotic pigs, and warthogs: (**A**) A heatmap of the SLA-2 MHCcluster. (**C**) A heatmap of the DQB-1 clustering. (**B**,**D**) Unrooted phylogenetic trees for MHC alleles with visualization of the MHC motifs showing the clustering of the SLA-2 and DQB1 alleles, respectively, with those of known SLA-2 and DQB1 alleles. The clusters obtained are circled (**B**,**D**).

**Table 1 vetsci-08-00180-t001:** The estimated dN/dS for the coding sequences of SLA-1 loci and the least-likelihood ratios for the three suid breeds studied.

Region	Population	Overall dN/dS Value (M0)	Log Likelihood Model 1 (M7, Neutral)	Log Likelihood Model 2 (M8, Selection)	Prop. of Sites under Positive Selection (p1) Estimated by M8	Estimated dN/dS Value under M8 for the Sites under Selection	LRT Statistic = −2 Delta (lnl)	Critical Value for the Chi-Square Test (df = 2) at 5% Alpha	*p*-Value (Selection vs. Neutral)	Significant at Alpha Level = 5%
*SLA-1* exon 2 (PBR, 80 codons)	All	0.59	−3799.06	−3709.20	0.01	3.36	179.71	5.99	0	Yes
	Exotic	0.43	−1234.51	−1227.72	0.04	2.52	13.58	5.99	0.0011	Yes
	Local	0.54	−2228.69	−2190.62	0.02	3.13	76.13	5.99	0	Yes
	Warthog	0.69	−1612.83	−1581.76	0.01	4.55	62.12	5.99	0.000001	Yes

**Table 2 vetsci-08-00180-t002:** The estimated dN/dS for the coding sequences of SLA-2 loci and the least-likelihood ratios for the three suid breeds studied.

*SLA−2* Loci	Population	Overall dN/dS Value (M0)	Log Likelihood Model 1 (M7, Neutral)	Log Likelihood Model 2 (M8, Selection)	Prop. of Sites under Positive Selection Estimated under M8	dN/dS Value under m8 for Sites under Selection	LRT Statistic = −2 delta (lnl)	*p*-Value (Selection vs. Neutral)	Significant at Alpha Level = 5%
All *SLA−2* exons	All	0.54	−7631.46	−7381.62	0.02	5.37	499.68	0	Yes
	Exotic	0.47	−4594.24	−4485.75	0.02	5.52	216.97	0	Yes
	Local	0.63	−4727.60	−4584.63	0.02	7.72	285.93	0	Yes
Exon 1 (non−PBR, 10 codons)	All	0.63	−83.33	−83.33	0.00	1.00	0.00	1	No
	Exotic	0.62	−50.61	−50.61	0.00	1.00	0.00	1	No
	Local	0.13	−59.18	−59.18	0.00	1.00	0.00	1	No
Exon 2 (PBR, 91 codons)	All	0.39	−2613.44	−2585.88	0.03	2.14	55.13	0.00001	Yes
	Exotic	0.34	−1460.17	−1450.36	0.03	2.46	19.63	0.00001	Yes
	Local	0.43	−1717.28	−1697.02	0.04	2.67	40.51	0.00001	Yes
Exon 3 (PBR, 92 codons)	All	0.63	−1837.16	−1760.56	0.01	6.80	153.19	0	Yes
	Exotic	0.54	−1152.89	−1111.09	0.02	8.59	83.59	0	Yes
	Local	0.94	−1205.34	−1163.72	0.03	9.07	83.25	0	Yes
Exon 4 (non−PBR, 92 codons)	All	0.17	−774.71	−774.52	0.05	1.36	0.38	0.825	No
	Exotic	0.12	−584.31	−584.31	0.00	1.00	0.00	1	No
	Local	0.16	−598.96	−598.95	0.09	1.00	0.00	0.998	No
Exon 5 (non−PBR, 37 codons)	All	0.79	−583.16	−581.87	0.34	1.65	2.57	0.276	No
	Exotic	0.87	−462.28	−461.19	0.28	1.96	2.18	0.336	No
	Local	0.81	−291.38	−290.44	0.26	2.64	1.89	0.387	No
Exon 6 (non−PBR, 11 codons)	All	4.02	−139.79	−129.77	0.20	12.09	20.04	0.0001	Yes
	Exotic	1.38	−122.15	−117.88	0.21	5.01	8.54	0.014	Yes
	Local	N/A	−71.28	−68.99	0.46	N/A	4.59	0.101	No
Exon 7 (non−PBR, 17 codons)	All	0.72	−244.99	−243.59	0.14	2.61	2.79	0.248	No
	Exotic	0.62	−180.57	−180.52	0.49	1.17	0.09	0.956	No
	Local	2.09	−130.49	−130.10	0.76	2.99	0.78	0.675	No

**Table 3 vetsci-08-00180-t003:** The estimated dN/dS analyses for the coding sequences of DQB1 loci and the least-likelihood ratios for the three suid breeds studied.

Region	Population	Overall dN/dS Value (M0)	Log Likelihood Model 1 (M7, Neutral)	Log Likelihood Model 2 (M8, Selection)	Prop. Of Sites under Positive Selection (p1) as Estimated under M8	Estimated dN/dS Value under M8 for the Sites under Selection	LRT Statistic = −2 Delta (lnl)	*p*-Value (Selection vs. Neutral)	Significant at Alpha Level = 5%
All *DQB1* exons	All	0.79	−2065.05	−1999.56	0.02	11.72	130.99	0	Yes
	Exotic	0.74	−1611.00	−1567.22	0.03	12.28	87.55	0	Yes
	Local	1.06	−1395.52	−1384.42	0.06	10.39	22.20	0.0001	Yes
	Warthog	0.43	−1055.07	−1055.07	0.00	1.00	0.00	1	No
Exon 1 (non−PBR, 26 codons)	All	0.22	−115.78	−115.78	0.00	1.00	0.00	1	No
	Exotic	0.45	−110.01	−110.01	0.00	1.00	0.00	1	No
	Local	0.44	−108.57	−108.57	0.00	1.00	0.00	1	No
	Warthog	0.00	−102.68	−102.68	0.00	1.00	0.00	1	No
Exon 2 (PBR, 90 codons)	All	0.62	−1092.77	−1069.99	0.01	7.13	45.57	0.0001	Yes
	Exotic	0.65	−778.98	−768.58	0.06	4.12	20.80	0.0001	Yes
	Local	0.71	−608.18	−606.12	0.24	2.53	4.13	0.1269	No
	Warthog	0.78	−366.08	−366.08	0.00	1.00	0.00	1	No
Exon 3 (non−PBR, 94 codons)	All	0.14	−479.76	−479.76	0.00	1.00	0.00	1	No
	Exotic	0.04	−427.40	−427.40	0.00	1.00	0.00	1	No
	Local	0.43	−414.48	−414.48	0.00	1.00	0.00	1	No
	Warthog	0.13	−393.72	−393.72	0.00	1.00	0.00	1	No
Exon 4 (non−PBR, 37 codons)	All	0.56	−162.40	−162.40	0.00	1.00	0.00	1	No
	Exotic	N/A	−147.19	−146.87	N/A	N/A	0.63	0.7308	No
	Local	0.56	−162.39	−162.39	0.00	1.00	0.00	1	No
	Warthog	1.00	−140.47	−140.47	0.00	1.04	0.00	1	No
Exon 5 (non−PBR, 6 codons)	All	N/A	−16.31	−16.01	N/A	N/A	0.62	0.7339	No
	Exotic	1.00	−12.01	−12.01	0.10	1.05	0.00	1	No
	Local	N/A	−16.31	−16.00	N/A	N/A	0.62	0.7340	No
	Warthog	1.00	−12.00	−12.00	0.00	1.04	0.00	1	No

**Table 4 vetsci-08-00180-t004:** Determination of *SLA-1* codons under positive selection by the Bayes empirical Bayes (BEB) approach.

Loci	Population	Model	dN/dS Value	Significance ^a^	Codons Predicted to Be under Positive Selection BEB Inference ^b^
*SLA-1* (exon 2)	All	M7 vs. M8	3.36	*p* < 0	6 **, 15 **, 53 **, 57 **, 58 **, 61 **, 64 **, 68 **
	Exotic	M7 vs. M8	2.52	*p* < 0.001	6 *, 15 *, 36 *, 53 **, 57 **, 58 **, 65 **, 68 **
	Local	M7 vs. M8	3.13	*p* < 0	15 **, 36 **, 53 **, 56 **, 57 **, 58 **, 59 **, 61 *, 68 **
	Warthog	M7 vs. M8	4.55	*p* < 0	15 **, 53 **, 58 *, 64 **, 65 **
SLA-2	All	M7 vs. M8	5.37	*p* < 0	11 *, 17 **, 20 **, 28 **, 34 **, 35 **, 56 **, 64 **, 69 *, 77 **, 78 **, 79 *, 80 **, 81 *, 84 *, 88 **, 90 **, 91 *, 92 **, 106 **, 125 **, 127 **, 128 **, 154 **, 158 **, 162 **, 163 **, 166 **, 167 **, 174 **, 178 **, 180 **, 181 *, 286 *, 288 **, 312 **, 322 **, 345 *, 350 *, 351 **
	Exotic	M7 vs. M8	5.52	*p* < 0	17 **, 20 **, 34 *, 35 **, 56* *, 74 **, 78 **, 79 *, 80 **, 81 **, 84 **, 88 **, 90 **, 92 *, 106 **, 108 **, 125 **, 127 *, 158 **, 162 *, 163 **, 166 **, 167 **, 174 **, 178 **, 181 **, 312 *, 330 *, 350 *, 351 **
	Local	M7 vs. M8	7.72	*p* < 0	11 *, 17 **, 20 **, 28 **, 34 **, 35 **, 55 *, 56 **, 64 **, 73 **, 77 **, 78 **, 80 **, 88 **, 90 **, 91 **, 92 **, 106 *, 113 **, 125 **, 154 **, 158 **, 162 **, 163 **, 166 **, 167 **, 174 **, 178**, 180**, 288 *, 322 **
DQB1	All	M7 vs. M8	11.72	*p* < 0	8 *, 31 **, 35 **, 40**, 47 *, 48 **, 50 **, 52 *, 59 **, 60 **, 65 *, 79 **, 85 *, 88 **, 93 **, 99 *, 204 *, 242 **
	Exotic	M7 vs. M8	12.28	*p* < 0	8 **, 31 **, 35 **, 40 *, 47 **, 48 **, 49 *, 50 **, 52 **, 59 **, 60 *, 65 *, 79 **, 85 *, 88 **, 89 *, 93 *, 204 **
	Local	M7 vs. M8	10.39	*p* < 0.00001	31 **, 35 **, 48 *, 52 *, 59 *, 60 *, 79 *, 88 *, 93 *, 113 *, 242 *

^a^ Significance was calculated by comparing the difference in the log likelihoods corresponding to the M7-M8 model. ^b^ Underlined are residues on the peptide-binding region (PBRs). * Posterior probability ≥ 0.95. ** Posterior probability ≥ 0.99.

**Table 5 vetsci-08-00180-t005:** Genetic diversity within SLA-1, SLA-2, and DQB1 loci among three suid breeds. N: Number of individuals; Ho: Mean observed heterozygosity; Hs: Mean gene diversity within population (expected heterozygosity); Ht: Total gene diversity; Dst: Gene diversity among samples; Dstp: Heterozygosity corrected Dst; Fis: Inbreeding coefficient per overall loci (Fis: 1-Ho/Hs); Dest: Measure of population differentiation.

Loci	Population	N	Ho	Hs	Ht	Htp	Dst	Dstp	Fst	Fstp	Fis	Dest
*SLA-1*	Exotic pigs	16	0.0341	0.0887	0.0887	NaN	0	NaN	0	NaN	0.6154	NaN
	Locally-adapted pigs	27	0.0778	0.1116	0.1116	NaN	0	NaN	0	NaN	0.3029	NaN
	Warthogs	16	0.0766	0.1637	0.1637	NaN	0	NaN	0	NaN	0.5323	NaN
	All included	59	0.0471	0.1133	0.1191	0.121	0.0058	0.0077	0.0485	0.0636	0.5843	0.0087
*SLA-2*	Exotic pigs	16	0.0203	0.0585	0.0585	NaN	0	NaN	0	NaN	0.653	NaN
	Locally-adapted pigs	25	0.0167	0.0633	0.0633	NaN	0	NaN	0	NaN	0.7367	NaN
	Warthogs	2	0.0038	0.0019	0.0019	NaN	0	NaN	0	NaN	−1	NaN
	All included	43	0.0102	0.046	0.0556	0.0588	0.0096	0.0128	0.173	0.2181	0.7786	0.0135
*DQB1*	Exotic pigs	16	0.0021	0.0197	0.0197	NaN	0	NaN	0	NaN	0.8911	NaN
	Locally-adapted pigs	21	0.0006	0.0099	0.0099	NaN	0	NaN	0	NaN	0.9428	NaN
	Warthogs	2	0.002	0.0152	0.0152	NaN	0	NaN	0	NaN	0.8696	NaN
	All included	39	0.0012	0.0147	0.0206	0.0225	0.0059	0.0079	0.2867	0.3489	0.9202	0.008

**Table 6 vetsci-08-00180-t006:** The DNA polymorphism measures for PBRs in *SLA-1*, *SLA-2*, and *DQB1*. Where N: The sample size; π: Nucleotide diversity; Hd: Haplotype (gene) diversity; θW: Watterson mutation estimator from variable sites, Eta: Total number of mutations; and θ_Eta_: Theta per site from Eta.

Loci	MHC	Pop	N	Π	Hd	θW	θEta	Eta
*SLA-1*	PBR (Exon 2)	All samples	60	0.12178 ± 0.00472	0.9973 ± 0.0015	0.13415 ± 0.03321	0.20751	198
		Locally-adapted pigs	27	0.10083 ± 0.00539	0.997 ± 0.004	0.0973 ± 0.02795	0.13146	127
		Exotic pigs	18	0.085 ± 0.00796	0.98 ± 0.010	0.08 ± 0.02747	0.11017	100
		Warthogs	16	0.11654 ± 0.00952	0.998 ± 0.08	0.12834 ± 0.04066	0.166	119
*SLA-2*	PBR (Exon 2)	All samples	44	0.1098 ± 0.00396	0.968 ± 0.011	0.09404 ± 0.02462	0.13149	158
		Locally-adapted pigs	25	0.09812 ± 0.00793	0.953 ± 0.021	0.08442 ± 0.02464	0.1135	121
		Exotic pigs	16	0.09582 ± 0.00464	0.944 ± 0.035	0.07725 ± 0.02459	0.09564	104
		Warthogs *	2	0.00494 ± 0.00151	0.667 ± 0.204	0.00404 ± 0.00324	0.00404	2
	PBR (Exon 3)	All samples	44	0.05975 ± 0.00311	0.925 ± 0.021	0.05956 ± 0.01593	0.07822	109
		Locally-adapted pigs	25	0.05030 ± 0.00434	0.878 ± 0.042	0.05015 ± 0.000406	0.06471	80
		Exotic pigs	16	0.06599 ± 0.00461	0.935 ± 0.035	0.05488 ± 0.01781	0.06658	74
		Warthogs *	2	0	0	0	0	0
*DQB1*	PBR (Exon 2)	All samples	40	0.03332 ± 0.00423	0.735 ± 0.055	0.04636 ± 0.01291	0.05085	68
		Locally-adapted pigs	21	0.02152 ± 0.00663	0.455 ± 0.095	0.03357 ± 0.01083	0.03529	41
		Exotic pigs	16	0,04146 ± 0.00442	0.905 ± 0.039	0.03863 ± 0.01293	0.04047	44
		Warthogs	2	0.01914 ± 0.00557	0.833 ± 0.222	0.01616 ± 0.00993	0.01616	8

* Warthog samples were almost identical and fewer in number.

## Data Availability

The datasets generated and analyzed for this study can be found in the European Nucleotide Archives (ENA) under the Project ID: PRJEB39824 with the following accession numbers: LR882502-LR882643.

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
