# Peer review of "Comparative Analysis of SLA-1, SLA-2, and DQB1 Genetic Diversity in Locally-Adapted Kenyan Pigs and Their Wild Relatives, Warthogs"

_vetsci, 2021, doi:10.3390/vetsci8090180_

Round 1
Reviewer 1 Report
Dear authors,
This study analyses diversity of three genes related to Swine leukocyte antigen (SLA), concretely SLA-1, SLA-2, and DQB-1 in three porcine breeds with different genetic background. The manuscript is well structured and written, the introduction provides sufficient background and include relevant references, the research design is appropriate, the methods are adequately described, the results are clearly presented, and the conclusions are supported by the results. However, some minor revisions are necessary:
- Line 63: written the name of genes in italics.
- Lines 109-116: From my point of view, in this paragraph the authors indicate ideas more specific to the conclusion, and information that should be included in the section "Material and methods".
- In the “Material and Methods” section, the authors should add a subsection “Statistical analysis”.
- Line 122: change “county” to “country”.
- Figure 1B: photographs of the three genetically different types of pigs analyzed at work would be interesting.
- Line 172: Why have the authors used GADPH as housekeeping and not another? More and more studies are being carried out showing that not all housekeeping works for all species and all types of samples. If there is a bibliography that indicates that this gene is the most suitable, please indicate it. If not, authors should use more than one housekeeping to ensure.
- Line 175: the inclusion of a table with the sequences of the all primers used would be convenient.
- Line 201: eliminate point before reference.
- Line 201: written reference correctly.
- Line 202: Add point after reference.
- Line 213: written correctly the reference.
- Line 392: written the name of genes in italics.
I hope these recommendations improve the quality of the manuscript.
Reviewer 2 Report
A brief summary
The article discusses the subject of the diversity of the molecular basis of the immunity of pigs kept in Kenya. It provides previously unavailable information on the genetic diversity of pigs reared in this country and supplements knowledge with valuable information that can be used in the prevention of pig disease, especially in the context of ASF.
Specific comments
Line 150: The material collected from the warthogs was blood or spleen, does this mean that only one type of tissue was collected from each individual? If so, is there no difference in the expression of the studied genes between these tissues when RNA is used, and whether the RNA produced in them can be compared? Is there any preliminary research done?
Linia 176: It would be helpful to include information on the sequences of the individual primers (a reference to supplementary table 1). Are the primer sequences for SLA-1 also derived from the previously cited source or were they designed by the authors?
Line 201: Incorrect citation
Linia 204: As in line 201
Linia 213: Duplicated citation, put after the dot, at the beginning of new sentence.
Linia 220: As in line 201
Line 261: In lines 147 and 148 it is mentioned that the material was collected from 27 locally adapted pigs and 16 exotic pigs for a total of 43 domestic pigs. However, this line states that 44 pigs were used. Is this number correct?
Line 266: As in line 201
Line 290: There is space missing after the parenthesis.
Line 303: The number of exotic pigs does not match the number given in line148, although the sum of exotic and locally adapted pigs coincides with the number of domestic pigs given in line 261. Are these numbers correct?
Line 339: Is this allele shared between warthogs and both breeds of pig, or with one breed only?
Line 382: This fragment is similar to the sentence beginning in line 378, but these values differ from each other. Are these the results obtained when warthogs were included in the analysis (which were not included in the previous section)? If so, this information should be included in the text to make it easier to understand.
Line 398: Table 8 is not in the text.
Line 402: As line 398
Line 430: 10 clusters are mentioned, but figure 6B shows 12 clusters.
Line 440: 6 clusters are mentioned but in figure 6D there is one allele not assigned to them. Shouldn't it be considered a separate cluster?
Line 491: There is no space before the square parenthesis.
Line 541: As in line 430
Line 552: The results obtained for DQB1 are not discussed.
Line 560: Does this mean that one individual express several MHC variants?
Line 587: Table 8 is not in the text.
Line 598: As in line 201
Table 2: There is „t” missing in „Significant”
Tables 1-5: Tables should be harmonized, in particular, Tables 1-3 which show the same type of results for different genes.
Figure 1: In the article, figure 1B (line 91) appears before figure 1A (line 120).
Figure 2: Allele detection circles are missing for CAB63936|SLA-1*01:01. Is this correct?
Figure 3: Allele detection circles are missing for AJ251829.1|SLA-2*03:0. Is this correct?
Figure 4: Allele detection circles are missing for NM_001113694.1|SLA-DQB1*03:0. Is this correct?
Figure 6: The colored cluster pointers are not used in Figures 6A and 6B as is the case in Figures 6C and 6D.
Figure 6: The clusters in figures 6A and 6C are indicated differently.
Supplementary Table 2 and 3: Table 3 (244) appears in the text before Table 2 (line 269).
Supplementary Tables 1-6: The tables are not named according to the guidelines.
Reviewer 3 Report
The resistance of pigs to stress and disease is of great concern to the pig industry. Analyzing the genetic diversity of stress resistance and disease resistance of pig breeds adapted to different climate characteristics can help pig breeders to select pigs with strong stress resistance and disease resistance in the future. In this design, 27 local-adapted pigs, 16 Landrace x Large White Crossed Exotic pigs and 16 Warthogs are used for blood and spleen samples. Genetic diversity analysis of SLA-1,SLA-2 and DQB1 genes in these samples was carried out and some conscious results were obtained. However, this paper has some shortcomings:
1 The warthog and the domestic pig are not of the same species. In this study, warthog was included in the genetic diversity comparison between Kenyan pigs and Landrace pigs, and the experimental design was incorrect.
2. The number of samples used in this study was too small for genetic analysis. It's usually need hundreds.
3. Phenotypic indexes such as some immunological indexes in the samples were not detected in this study. The correlation analysis between phenotypes such as immune indexes and genetic diversity can obtain more scientific results.
4 SLA-1, SLA-2 and DQB1 genes are some of the genes related to disease resistance in pigs. Comparison of these three genes cannot reflect the disease resistance characteristics of native Kenyan pigs.
Reviewer 4 Report
This manuscript has very interesting and valuable findings regarding leukocyte antigen loci in domestic pigs and warthogs in Kenya. This study has tremendous value for informing and enhancing animal agriculture; however, there are some issues with this work that I indicate below that should be addressed prior to publication.
- The authors discuss but do not present basic descriptive statistics of alleles for the local pigs, exotic pigs, and warthogs (e.g. lines 480-481). A major focus of the manuscript is to demonstrate higher allele diversity among locally adapted pigs and warthogs, the authors should present a summary of these data in the context of each population. Ideally for each population at each locus this would include number or frequency of alleles, observed and expected heterozygosity, and Hardy-Weinberg proportions.
- Details about sample collection is missing important details. How were samples taken from the animals, particularly the warthogs? The authors should indicate if these are captive or free-roaming animals, and if the samples were taken ante or post mortem.
- Parameters for PHASE are missing: MCMC samples, burn-in, frequency of samples drawn, number of repetitions.
- Warthog values are missing from table 2.
Round 2
Reviewer 3 Report
I think the author has answered the reviewer's opinion well. Some statements in this article need further modification.